# Comparison of the Malignant Predictors in Intrahepatic and Extrahepatic Intraductal Papillary Neoplasm of the Bile Duct

**DOI:** 10.3390/jcm11071985

**Published:** 2022-04-02

**Authors:** Sung Yong Han, Dong Uk Kim, Hyeong Seok Nam, Dae Hwan Kang, Sung Ill Jang, Dong Ki Lee, Dong Woo Shin, Kwang Bum Cho, Min Jae Yang, Jae Chul Hwang, Jin Hong Kim, Hoonsub So, Sung Jo Bang, Min Je Sung, Chang-Il Kwon, Dong Wook Lee, Chang-Min Cho, Jae Hee Cho

**Affiliations:** 1Department of Internal Medicine, Biomedical Research Institute, Pusan National University Hospital, Pusan National University School of Medicine, Busan 49241, Korea; mirsaint@hanmail.net; 2Department of Internal Medicine, Research Institute for Convergence of Biomedical Science and Technology, Pusan National University Yangsan Hospital, Pusan National University School of Medicine, Yangsan 50612, Korea; skyace27@hanmail.net (H.S.N.); sulsulpul@naver.com (D.H.K.); 3Department of Internal Medicine, Gangnam Severance Hospital, Yonsei University College of Medicine, Seoul 06230, Korea; aerojsi88@gmail.com (S.I.J.); dklee@yuhs.ac (D.K.L.); 4Department of Internal Medicine, Dongsan Medical Center, Keimyung University School of Medicine, Daegu 42601, Korea; delight0618@naver.com (D.W.S.); chokb@dsmc.or.kr (K.B.C.); 5Department of Gastroenterology, Ajou University School of Medicine, Suwon 16499, Korea; creator1999@hanmail.net (M.J.Y.); cath07@naver.com (J.C.H.); jinhkim@ajou.ac.kr (J.H.K.); 6Department of Internal Medicine, Ulsan University Hospital, University of Ulsan College of Medicine, Ulsan 44033, Korea; hoon3112@gmail.com (H.S.); sjbang@uuh.ulsan.kr (S.J.B.); 7Digestive Disease Center, CHA Bundang Medical Center, CHA University School of Medicine, Seongnam 13497, Korea; mj1744@hanmail.net (M.J.S.); mdkwon@naver.com (C.-I.K.); 8Department of Internal Medicine, Kyungpook National University School of Medicine, Daegu 42601, Korea; storm5333@naver.com (D.W.L.); changmincho@gmail.com (C.-M.C.)

**Keywords:** intraductal papillary neoplasm of the bile duct, prognosis, natural course, cholangiocarcinoma, predictor

## Abstract

Background: Intraductal papillary neoplasm of the bile duct (IPNB) is a precancerous lesion of cholangiocarcinoma, for which surgical resection is the most effective treatment. We evaluated the predictors of malignancy in IPNB according to anatomical location and the prognosis without surgery. Methods: A total of 196 IPNB patients who underwent pathologic confirmation by surgical resection or endoscopic retrograde cholangiography or percutaneous transhepatic cholangioscopic biopsy were included. Clinicopathological findings of IPNB with invasive carcinoma or mucosal dysplasia were analyzed according to anatomical location. Results: Of the 116 patients with intrahepatic IPNB (I-IPNB) and 80 patients with extrahepatic IPNB (E-IPNB), 62 (53.4%) and 61 (76.3%) were diagnosed with invasive carcinoma, respectively. Multivariate analysis revealed that mural nodule > 12 mm (*p* = 0.043) in I-IPNB and enhancement of mural nodule (*p* = 0.044) in E-IPNB were predictive factors for malignancy. For pathologic discrepancy before and after surgery, IPNB has a 71.2% sensitivity and 82.3% specificity. In the non-surgical IPNB group, composed of nine I-IPNB and seven E-IPNB patients, 43.7% progressed to IPNB with invasive carcinoma within 876 days. Conclusions: E-IPNB has a higher rate of malignancy than I-IPNB. The predictive factor for malignancy is mural nodule > 12 mm in I-IPNB and mural nodule enhancement in E-IPNB.

## 1. Introduction

Cholangiocarcinoma (CCA) has a poor prognosis, and its incidence is increasing worldwide [1]. Most patients with CCA are diagnosed at advanced stages with low cure rates, even after aggressive therapy [2]. Early diagnosis or detection of precancerous lesions is important in improving patient survival. The World Health Organization classified two premalignant lesions in CCA: (1) biliary intraepithelial neoplasia (Bil-IN) and (2) intraductal papillary neoplasm of the bile duct (IPNB) [3]. Bil-IN is difficult to diagnose with imaging tests before surgery and is usually diagnosed pathologically after surgical resection. IPNB is the only macroscopic precancerous lesion of CCA that can be diagnosed preoperatively.

CCA is a heterogeneous disease group that includes intrahepatic, extrahepatic, and perihilar types. Each CCA has a different epidemiology, physiology, prognosis, and treatment strategy [1]. IPNB is also known to show differences in intrahepatic and extrahepatic lesions. Intrahepatic IPNB (I-IPNB) has a better prognosis, and the rate of malignancy is lower than that of extrahepatic IPNB (E-IPNB) [4]. The treatment of choice for I-IPNB is surgical resection, with a recurrence rate of 24–32% and a 5-year survival rate of 68–80% [4,5]. However, surgical resection may not be suitable for some patients with advanced age, many comorbidities, and multifocal lesions. On the other hand, E-IPNB has a high malignancy rate and may cause cholangitis. Therefore, surgical resection should be performed if possible. The natural course of non-operative IPNB is unknown. Currently, there are no known factors predicting malignancy, but if these factors could be identified, they would be useful in deciding the need for surgical treatment for I-IPNB. Recently, some cases of local treatment, such as radiofrequency ablation (RFA), photodynamic therapy (PDT), and argon plasma coagulation (APC), have been reported in patients who are difficult to treat [6,7,8,9]. Although long-term results have not been obtained, these local treatments could be performed in patients with a low risk for malignancy and in those who are not suitable for surgery.

In this study, we evaluated the predictors of malignancy in E-IPNB and I-IPNB. We also investigated the natural course of benign IPNB without surgical resection.

## 2. Patients and Methods

### 2.1. Patients

A total of 196 patients with IPNB who underwent pathologic confirmation by surgical resection or endoscopic retrograde cholangiopancreatography (ERCP) or percutaneous transhepatic cholangioscopy (PTCS) biopsy were retrospectively reviewed in eight medical institutions between January 2011 and September 2021. The inclusion criteria were as follows: (1) pathologic confirmation of IPNB, and (2) patients with abdominal imaging (enhanced computed tomography (CT) or magnetic resonance imaging (MRI)) before surgery or biopsy performed by ERCP, or PTCS. IPNB associated with invasive carcinoma has stromal invasion in IPNB. To evaluate the natural course of IPNB, we additionally evaluated the progression to invasive carcinoma in patients who did not undergo surgery and had had an image scan at least more than six months after pathologic confirmation of IPNB. This was also to avoid the underestimation of malignancy. This study was conducted in accordance with the ethical guidelines of the Declaration of Helsinki (revised in 2013), and the study protocol was approved by the Institutional Review Board of our Hospital (no. 2010-014-096).

### 2.2. Measurement of Parameters

We analyzed symptoms, laboratory findings, radiological information, and clinical information, such as age, sex, body mass index (BMI), and comorbidities (diabetes, hypertension, hepatitis, liver cirrhosis). Laboratory data included white blood cell count (WBC), alanine aminotransferase (ALT), total bilirubin (TB), alkaline phosphatase (ALP), gamma-glutamyl peptidase (GGT), C-reactive protein (CRP), carcinoembryonic antigen (CEA), and carbohydrate antigen (CA19-9). Cutoff values for TB, CEA, and CA19-9 were defined as 3 mg/dL, 5 ng/mL, and 37 U/mL, respectively, and patients with abnormal values were investigated. Radiologic findings included tumor location (hilar lesions were included in extrahepatic lesions), mural nodule (present or absent, size and enhancement), intrahepatic duct (IHD) stones, focal atrophy, and lymph node enlargement. The duct size was measured, along with the bile duct of the tumor. Abrupt change of the bile duct was defined as a decrease of >50% in bile duct size before and after the tumor. The cutoff value for mural nodules was defined by the receiver operating characteristic (ROC) curve.

In non-operative IPNB patients, the periods of malignant change and follow-up were checked. If a significant change was found in the imaging study, a biopsy was performed to confirm the malignant change. The occurrence of cholangitis was assessed during the follow-up period. However, this was limited to hospitalized patients with cholangitis.

### 2.3. Statistical Analysis

Statistical analysis was performed using the SPSS software (version 21.0, IBM Corp., Armonk, NY, USA). Categorical data were expressed as frequency and percentage, and between-group differences were evaluated using the chi-square test. Continuous data were expressed as mean ± standard deviation (SD), with between-group differences evaluated using an independent Student’s *t*-test. Statistical significance was set at *p* < 0.05. Univariate and multivariate analyses were conducted to identify the predictors of malignancy. Variables with a *p*-value < 0.05 in the univariate analysis were included in the multivariate analysis.

## 3. Results

### 3.1. Baseline Characteristics 

A total of 196 patients with IPNB were enrolled in the study. There were 116 patients with I-IPNB and 80 patients with E-IPNB. Among the patients with I-IPNB, 62 (53.4%) had invasive carcinoma, while among the patients with E-IPNB, 61 (76.3%) had invasive carcinoma. Table 1 shows the clinical and radiologic characteristics of the patients with I-IPNB and E-IPNB. In the I-IPNB group, the age, sex ratio, BMI, diabetes, hypertension, liver cirrhosis, and initial symptoms were not different between IPNB with invasive carcinoma (malignant group) and IPNB with mucosal dysplasia (benign group). Viral hepatitis was more common in the benign group than in the malignant group (1.6% vs. 13.0%, *p* = 0.033). Most laboratory findings, including WBC, ALT, TB, ALP, GGT, and CRP, were not significantly different between the two groups. However, CEA and CA-19-9 were significantly higher in the malignant group than in the benign group (CEA: 27.8% vs. 7.5%, *p* = 0.013 and CA19-9: 52.5% vs. 21.7%, *p* = 0.001). In the radiologic findings, the mean size of mural nodule (18.5 vs. 10.9 mm, *p* = 0.013), mural nodule > 12 mm (69.0% vs. 33.3%, *p* = 0.012), and lymph node enlargement (16.1% vs. 3.7%, *p* = 0.028) were significantly higher in the malignant group than in the benign group. However, the presence of IHD stone (11.3% vs. 40.7%, *p* < 0.001) was significantly lower in the malignant group.

In the E-IPNB group, over 3 mg/dL of TB (44.3% vs. 15.8%, *p* = 0.025), >37 IU/L of CA19-9 (54.4% vs. 22.2%, *p* = 0.017), and enhancement of mural nodules (90.6% vs. 55.6%, *p* = 0.013) were significantly higher in the malignant group than in the benign group.

### 3.2. Predictors of Malignancy

To evaluate the predictors of malignancy, logistic regression analysis was performed using the factors that were clinically or statistically meaningful in I-IPNB and E-IPNB (Table 2 and Table 3). In I-IPNB, CEA > 5 U/mL (risk ratio (RR): 4.74, 95% confidence interval (CI): 1.27–17.73, *p* = 0.021), CA19-9 > 37 IU/L (RR: 3.99, 95% CI: 1.68–9.49, *p* = 0.002), mural nodule > 12 mm (RR: 4.44, 95% CI: 1.33–14.77, *p* = 0.015), IHD stones (RR: 0.018, 95% CI: 0.07–0.48, *p* = 0.001), and LN enlargement (RR: 5.00, 95% CI: 1.04–23.94, *p* = 0.044) were significant predictors of malignancy. In the multivariate analysis, mural nodule > 12 mm (RR: 5.33, 95% CI: 1.05–26.89, *p* = 0.043) was the only significant predictor of malignancy of I-IPNB. On the other hand, in E-IPNB, TB > 3 mg/dl (RR: 4.23, 95% CI: 1.11–16.05, *p* = 0.034), CA19-9 > 37 U/L (RR: 4.17, 95% CI: 1.22–14.2, *p* = 0.023), and enhancement of mural nodules (RR: 7.73, 95% CI: 1.31–45.51, *p* = 0.024) were identified as predictors of malignancy based on the univariate analysis. However, in the multivariate analysis, enhancement of mural nodules (RR: 19.08, 95% CI: 1.08–335.5, *p* = 0.044) was the only significant predictor of malignancy.

### 3.3. Pathologic Discrepancy

Table 4 shows the pathologic discrepancies of IPNB. A total of 85 patients, 27 with I-IPNB and 58 with E-IPNB, who underwent biopsy before surgical resection, were included in the analysis. I-IPNB had a sensitivity of 76.9%, specificity of 100%, positive predictive value of 100%, and negative predictive value of 71.4%. E-IPNB had a sensitivity of 69.5%, specificity of 83.3%, positive predictive value of 94.1%, and negative predictive value of 41.6%.

### 3.4. Prognosis of Non-Operative IPNB Patients

Table 5 shows the prognosis of non-operative I-IPNB and E-IPNB patients. Nine I-IPNB patients who did not undergo surgery were observed during a median follow-up period of 1091 days (range: 361–2393 days). Among them, five patients (55.5%) were diagnosed with malignancy with a median malignant transformation period of 876 days (range: 569–1590 days), and three patients (33.3%) were hospitalized due to cholangitis. Only two patients (22.2%) did not experience admission or malignant change during the follow-up period. Seven E-IPNB patients who did not undergo surgery were also observed during a median follow-up period of 1200 days (range: 324–1285 days). Among them, two patients (28.5%) developed malignancy with a median malignant transformation period of 937.5 days (range: 870–1005 days). Four patients (57.0%) were hospitalized for cholangitis, while three patients (42.8%) were not admitted due to cholangitis or any malignant change.

Table 6 shows the clinical information, including initial and last follow-up of CEA, CA19-9, location of the tumor, mural nodule, enhancement, pathologic result, follow-up period, presence of cholangitis, and presence of malignant change, of non-operative IPNB patients.

## 4. Discussion

I-IPNB had a significantly lower malignancy rate than E-IPNB. In the multivariate analysis, a mural nodule > 12 mm was the only identified predictor of I-IPNB, while enhancement of mural nodules was the identified predictor of E-IPNB. In terms of pathologic discrepancy between preoperative biopsy and surgical histologic results, there was a low negative predictive value in E-IPNB (41.6%) and a relatively high one in I-IPNB (71.4%). A total of 16 patients diagnosed with IPNB by biopsy did not undergo surgical resection, and follow-up observation was performed. Among them, seven patients (43.7%) were diagnosed with a malignancy within three years. Only five patients (31.2%) did not experience admission due to cholangitis or change in malignancy during the follow-up period.

Mural nodule > 12 mm and enhancement of mural nodule are predictors of malignancy in I-IPNB and E-IPNB, respectively. These results are consistent with the findings in intraductal papillary mucinous neoplasm (IPMN). Similarities between biliary and pancreatic premalignant lesions can be explained by the fact that the embryological development of the bile duct and the main pancreatic duct originates from the hepatic diverticulum in the foregut mesoderm [10]. IPNB displays similarities to main duct IPMN, which is more aggressive than branch duct IPMN [11]. In the 2012 consensus, there were five worrisome and three high-risk stigmata [12]. Therefore, we used these predictors of malignant IPMN, such as obstructive jaundice, duct size, mural nodule, and abrupt size change, as potential predictors of malignancy of IPNB. Similar to IPMN, IPNB with large mural nodules and enhancement of mural nodules are more likely to be malignant. CA19-9 is not an independent predictor in our results; however, a recent study showed that CA19-9 is a prognostic marker of I-IPNB with high-grade dysplasia and carcinoma [13]. Therefore, despite our results, it should be noted that CA19-9 is always important as prognostic markers of the biliary tract cancer. 

Unlike IPMN, IPNB requires surgical resection in all patients [11,14]. The prognosis after surgical resection slightly differs depending on the pathological results or invasiveness and location of the tumor, but it is known to be very good with a 5-year survival rate of 68–80% [4,15]. However, in some patients, surgery may be risky because of advanced age or the presence of comorbidities. In patients with multifocal lesions, it may be necessary to resect only the lesion that is most suspicious for malignancy. Therefore, for these patients, careful decision-making is needed to reduce the risks. For E-IPNB, surgery should be performed if possible. In some studies, including ours, E-IPNB has been shown to have a high rate of malignancy [5]. The pathologic discrepancy results also support the need for surgery in E-IPNB. In E-IPNB, the negative predictive value was 41.6%, indicating that around 60% of patients who received a non-malignancy result actually had a malignancy. However, I-IPNB had a higher rate of negative predictive value (71.4%). This difference may be caused by the biopsy method. In E-IPNB, biopsy was performed using ERCP, through a fluoroscopic image. Therefore, accurate targeting biopsy is difficult. In I-IPNB, biopsy was performed through PTCS, making it possible to conduct an accurate biopsy while directly inspecting the lesion. However, PTCS biopsy was not performed in many cases because of the difficulty of the procedure. In addition, some reports have shown that the pathological characteristics between I-IPNB and E-IPNB are different. Recently, IPNB has been pathologically divided into type 1 and type 2. Type 2 is more commonly found in the E-IPNB and has unfavorable postoperative outcomes [16,17,18]. Type 1 is thought to be similar to IPMN, while type 2 is slightly different [18]. Mutations expressed according to type 1/2 have also been reported to be different. In type 1, KRAS, GNAS, and RNF43 mutations were identified and reported to be similar to the intestinal subtype of IPMN. In type 2, mutations in TP53, SMAD4, and PIK3CA were identified [19]. I-IPNB also requires surgery when diagnosed. I-IPNB has a good prognosis and a low malignancy rate. Therefore, in patients at high risk for surgery, if possible, performing a biopsy through PTCS, and determining whether there is malignancy and the type of mutations will be helpful in deciding whether to push through with the surgery.

We report the natural course of patients who did not undergo surgery. To the best of our knowledge, we have reported on the largest number of patients among the articles discussing the natural course of IPNB. During the span of approximately three years, among nine patients with I-IPNB, five were diagnosed with malignancy and three were admitted to the hospital because of cholangitis. Of the seven patients with E-IPNB who did not undergo surgery, two developed malignancy, and four were hospitalized for cholangitis. Only three patients were not admitted due to cholangitis or malignant transformation during the follow-up period. According to the results of our analysis of the natural course, 22.2% of the patients had no complications during the three years. The incidence of cholangitis was high in E-IPNB, while the rate of malignant change was high in I-IPNB. There is no established treatment strategy for patients who are difficult to treat, but additional treatment should be considered. As mentioned above, many methods have been recently reported for local treatment, including PDT, RFA, and APC, in IPNB [6,7,8,9]. Although there is still a need for studies on the safety and effectiveness of these local treatments, they are thought to be options for patients with non-malignant I-IPNB for whom it will be difficult to perform surgical treatment.

Our study had some limitations. First, the number of enrolled patients was relatively small compared to other articles, and the number of patients in whom the natural course was observed was too small to draw conclusions from. Nevertheless, it is meaningful because there are few articles on the natural course of IPNB. Second, we did not analyze the pathologic subtypes of IPNB. Our study is related to preoperative characteristics, and it was difficult to accurately identify the subtype by histological biopsy before surgery. In addition, IPNB classification types 1 or 2 have not yet been fully established. More studies should be published on IPNB, including its natural course, histological results, and future mutations.

In conclusion, mural nodule > 12 mm was identified as a predictor of malignancy in I-IPNB, and enhancement of mural nodules was a predictor of malignancy in E-IPNB. In addition, 43.7% of non-operative patients with mucosal dysplasia had malignant transformation within three years. We suggest that follow-up observation is performed only for patients who are expected not to have malignancy and where it is considered difficult for them to undergo surgical treatment.

## Figures and Tables

**Table 1 jcm-11-01985-t001:** Clinical and radiologic characteristics for predicting malignancy in univariate analysis.

	Intrahepatic IPNB (*N* = 116)			Extrahepatic IPNB (*N* = 80)		
	IPNB with Invasive Carcinoma (*N* = 62)	IPNB with Mucosal Dysplasia (*N* = 54)	*p*-Value	IPNB with Invasive Carcinoma (*N* = 61)	IPNB with Mucosal Dysplasia (*N* = 19)	*p*-Value
Age (median ± SD)	70.5 ± 8.4	67.8 ± 10.0	0.114	69.1 ± 9.3	69.9 ± 11.8	
Sex (male (%))	34 (54.8)	32 (59.3)	0.635	41 (67.2)	11 (57.9)	0.463
BMI	23.5 ± 3.6	23.5 ± 3.0	0.926	23.3 ± 3.2	23.1 ± 4.1	0.860
Diabetes	12 (19.4)	13 (24.1)	0.542	9 (14.8)	4 (21.1)	0.522
Hypertension	23 (37.1)	24 (44.4)	0.426	23 (37.7)	8 (42.1)	0.735
Symptoms			0.271			0.456
Abdominal pain	13 (21.7)	19 (35.2)		16 (26.2)	4 (21.1)	
Jaundice	7 (11.7)	4 (7.4)		28 (45.9)	3 (15.8)	
Fever	7 (11.7)	3 (5.6)		1 (1.6)	3 (15.8)	
etc.	33 (55.0)	28 (51.9)		16 (26.3)	9 (46.4)	
Hepatitis (HBV/HCV)	0 (0)/1 (1.6)	3 (5.6)/4 (7.4)	0.033 *	4 (6.6)/0 (0)	1 (5.3)/0 (0)	0.841
Liver cirrhosis	5 (85.2)	2 (3.7)	0.319	2 (3.3)	1 (5.3)	0.695
Laboratory Finding						
WBC (10^3^)	7.2 ± 2.7	10.0 ± 15.2	0.155	7.9 ± 3.6	7.4 ± 12.6	0.611
ALT	71.3 ± 134.5	79.5 ± 141.2	0.751	107.4 ± 159.4	87.1 ± 141.1	0.621
TB	1.73 ± 3.40	1.32 ± 1.61	0.420	4.41 ± 5.76	1.83 ± 2.85	0.065
>3 mg/dL	8 (12.9)	4 (7.4)	0.337	27 (44.3)	3 (15.8)	0.025 *
ALP	196.9 ± 162.6	168.9 ± 187.2	0.392	377.5 ± 339.4	304.9 ± 325.6	0.414
GGT	276.9 ± 354.4	250.4 ± 404.6	0.392	480.1 ± 504.6	363.5 ± 450.7	0.385
CRP	2.55 ± 4.55	4.02 ± 7.41	0.212	3.47 ± 4.83	3.63 ± 6.42	0.913
CEA	170.1 ± 834.6	2.6 ± 1.7	0.208	5.1 ± 10.5	3.0 ± 1.6	0.419
>5 U/mL	15 (27.8)	3 (7.5)	0.013 *	9 (18.4)	3 (16.7)	0.875
CA19-9	1161.3 ± 4068.6	92.3 ± 363.3	0.079	764.9 ± 3371.3	4341.7 ± 18,321.8	0.160
>37 U/L	31 (52.5)	10 (21.7)	0.001 *	31 (54.4)	4 (22.2)	0.017 *
Multifocal	10 (16.1)	4 (7.4)	0.153			
Duct size (mm)	9.8 ± 4.6	9.7 ± 7.5	0.909	12.6 ± 6.6	10.3 ± 5.9	0.188
Mural nodule (*N*)	29 (46.8)	21 (38.9)	0.397	32 (52.5)	9 (47.4)	0.703
Size (mm)	18.5 ± 11.8	10.9 ± 7.1	0.013 *	13.2 ± 7.0	9.5 ± 4.6	0.153
>12 mm	20/29 (69.0)	7/21 (33.3)	0.012 *	15/32 (46.9)	3/9 (33.3)	0.482
Enhanced	22/29 (75.9)	13/21 (61.9)	0.297	29/32 (90.6)	5/9 (55.6)	0.013 *
Abrupt change of the bile duct (*N*)	18 (30.5)	8 (19.5)	0.222	18 (29.5)	2 (11.1)	0.118
IHD stone	7 (11.3)	22 (40.7)	<0.001 *			
Focal atrophy	15 (24.2)	19 (35.2)	0.198			
LN enlargement	10 (16.1)	2 (3.7)	0.028 *	12 (19.7)	4 (21.1)	0.897

* *p*-Value < 0.005.

**Table 2 jcm-11-01985-t002:** Risk ratio and logistic regression analysis in intrahepatic IPNB.

	RR (95% CI)	*p*-Value	Logistic Regression	*p*-Value
Hepatitis	0.31 (0.09–1.09)	0.067	78,823,444 (0–)	0.999
CEA > 5 U/mL	4.74 (1.27–17.73)	0.021 *	0.98 (0.05–19.19)	0.992
CA19-9 > 37 U/L	3.99 (1.68–9.49)	0.002 *	1.13 (0.11–11.15)	0.916
Multifocal lesion	2.40 (0.71–8.17)	0.160	3.45 (0.29–41.02)	0.327
Mural nodule > 12 mm	4.44 (1.33–14.77)	0.015 *	5.33 (1.05–26.89)	0.043 *
Enhanced mural nodule	1.93 (0.57–6.58)	0.291	2.80 (0.51–15.2)	0.233
Abrupt change of the bile duct	1.81 (0.70–4.68)	0.221	3.94 (0.37–41.79)	0.254
IHD stone	0.18 (0.07–0.48)	0.001 *	0.54 (0.04–8.28)	0.655
LN enlargement	5.00 (1.04–23.94)	0.044 *	3.84 (0.15–96.70)	0.413

* *p*-Value < 0.005.

**Table 3 jcm-11-01985-t003:** Risk ratio and logistic regression analysis in extrahepatic IPNB.

	RR (95% CI)	*p*-Value	Logistic Regression	*p*-Value
TB > 3 mg/dL	4.23 (1.11–16.05)	0.034 *	0.29 (0.18–4.89)	0.396
CA19-9 > 37 U/L	4.17 (1.22–14.2)	0.023 *	10.59 (0.87–128.12)	0.063
Mural nodule > 12 mm	1.77 (0.37–8.31)	0.473	1.96 (0.12–29.85)	0.628
Enhanced mural nodule	7.73 (1.31–45.51)	0.024 *	19.08 (1.08–335.5)	0.044 *
Abrupt change of the bile duct	3.34 (0.69–16.09)	0.131	3.61 (0.23–56.71)	0.360

* *p*-Value < 0.005.

**Table 4 jcm-11-01985-t004:** Pathologic discrepancy pre- and post-operation.

Pre operation → Post Operation	Total * (*n* = 85)	Intrahepatic IPNB (*n* = 27)	Extrahepatic lPNB (*n* = 58)
carcinoma → mucosal dysplasia	42	10	32
Mucosal dysplasia → carcinoma	17	3	14
carcinoma → Mucosal dysplasia	2	0	2
Mucosal dysplasia → Mucosal dysplasia	24	14	10
Sensitivity	71.2%	76.9%	69.5%
Specificity	92.3%	100%	83.3%
Positive predictive value	95.5%	100%	94.1%
Negative predictive value	58.5%	71.4%	41.6%

* Enrolled patients who had both pre- and post-operation pathology result. (Defined: carcinoma **→** positive result, mucosal dysplasia **→** negative result).

**Table 5 jcm-11-01985-t005:** prognosis of non-operative IPNB patients.

Variables	Total Patients (*N* = 16)	
	Intrahepatic (*N* = 9)	Extrahepatic (*N* = 7)
Follow up period (median, range)	1091 days (361–2393)	1200 days (324–1285)
Malignant transformation (*n*, (%))	5 (55.5)	2 (28.5)
Malignant transformation period (median, range)	876 days (569–1590)	937.5 days (870–1005)
Admission due to cholangitis during follow up period (*n*, (%))	3 (33.3)	4 (57.0)
No admission & malignant change	2 (22.2)	3 (42.8)

**Table 6 jcm-11-01985-t006:** clinical information of patients with IPNB without surgery.

No	Sex	Age	CEA		CA19-9		Location		Mural Nodule (Enhanced)		Initial Pathology	Follow-Up Period (Days)	Cholangitis (Frequency)	Malignant Change (Days)
			Initial	Last f.u	Initial	Last f.u	Initial	Last f.u	Initial	Last f.u				
1	male	67	2.3	2.3	4.3	4.4	intra	intra	-	39 (yes)	LGD	1091	No	Yes (1091)
2	male	72	4.7	4.2	40.2	351.1	intra	intra & extra	5 (no)	20 (yes)	HGD	1670	No	Yes (876)
3	female	75	0.6	0.9	16.6	35.3	intra	intra	5 (no)	9 (yes)	LGD	810	No	Yes (690)
4	male	71	3.1	3.6	77.1	113.2	intra	intra	14 (no)	56 (yes)	LGD	2190	No	Yes (1590)
5	male	71	2.5	6.0	48.6	58.7	intra	intra	22 (yes)	42 (yes)	LGD	805	Yes (#3)	Yes (569)
6	male	89	3.4	2.6	16.4	7.4	intra	intra	9 (no)	9 (no)	HGD	1110	Yes (#2)	No
7	male	83	0.5	2.6	700	24.5	intra	intra	-	-	LGD	361	Yes (#1)	No
8	male	66	2.0		0.8		intra	intra	-	-	LGD	2393	No	No
9	male	80					intra	intra & extra	4 (no)	-	LGD	520	No	No
10	male	76	1.0	6.2	8.3	113.0	extra	extra	-	-	HGD	1285	Yes (#6)	Yes (1005)
11	male	73	3.6	5.1	24.5	74.8	extra	extra	6 (yes)	8 (yes)	HGD	1200	Yes (#3)	Yes (870)
12	male	80	3.6	4.5	19.5	309.0	perihilar	perihilar & extra	15 (yes)	18 (yes)	LGD	1275	Yes (#4)	No
13	male	82	3.4	3.4	12.9	10.5	extra	extra	8 (no)	8 (no)	HGD	455	Yes (#1)	No
14	male	74	4.5	4.2	6.2	6.5	perihilar	perihilar & extra	-	3 (no)	LGD	324	No	No
15	male	75	2.8		27.8		extra	extra	15 (yes)	-	HGD	1253	No	No
16	female	84	3.3	3.2	77.3	27.2	extra	extra	-	-	HGD	546	No	No

## Data Availability

All relevant data contained within the article.

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
