# Peer review of "Comparison of the Malignant Predictors in Intrahepatic and Extrahepatic Intraductal Papillary Neoplasm of the Bile Duct"

_jcm, 2022, doi:10.3390/jcm11071985_

Round 1

Reviewer 1 Report

I would like to congratulate the authors for this paper showing malignant predictors in intra and extrahepatic  intraductal papilary neoplasm(IPNB).

-Mucin hypersecretion was shown to be more freqent in intrahepatic  IPNB then extrahepatic IPBN and where associated with in situ carcinome or minimally invasive carcinoma,while the lack of mucine hypersecretionwas associated with more invasive carcinoma.Is this information available in patients included in this study?

- would like to ask the authors if any molecular analysis have been  done or planned ?

Some datas suggests that IPBN develomment follows adenoma/adenocarcinoma sequence initiated by activation of kras and tp53 ,followed by loss of SMAD 4   and other molecular alterations in later stages.Thus, knowing molecular charactersation of IPBN could  help in clinical decision  extent of surgery for example).What is your opinion about ?

Author Response

I would like to congratulate the authors for this paper showing malignant predictors in intra and extrahepatic  intraductal papilary neoplasm(IPNB).

-Mucin hypersecretion was shown to be more freqent in intrahepatic  IPNB then extrahepatic IPBN and where associated with in situ carcinome or minimally invasive carcinoma, while the lack of mucine hypersecretion was associated with more invasive carcinoma. Is this information available in patients included in this study?

Reply: Thank you for your comment. We agree with your opinion as mucin production is related to invasiveness. However, we focused on malignancy itself rather than invasiveness. And our study is based on radiological images and laboratory findings. Therefore, mucin production is too difficult to predict using radiological images alone. Bile duct diameter is one of the variables used to predict mucin production. Hence, we included bile duct diameter as a variable. However, there was no significant difference between dysplasia and malignancy.

- would like to ask the authors if any molecular analysis have been done or planned ?

Some datas suggests that IPBN develomment follows adenoma/adenocarcinoma sequence initiated by activation of kras and tp53 ,followed by loss of SMAD 4  and other molecular alterations in later stages.Thus, knowing molecular charactersation of IPBN could  help in clinical decision  extent of surgery for example).What is your opinion about ?

Reply: Thank you for your comment. We did not include molecular analysis. However, I agree with your opinion that molecular alteration will be one of the important clues in determining the extent of surgery. As you mentioned, pathologically normal tissues adjacent to the tumor may harbor early molecular changes such as TP53 and SMAD4. Molecular profiling can also be used to determine the safe resection margins for surgery. If early molecular changes are detected at the margin of surgical resection, we believe that the likelihood of recurrence is high. A lot of related research needs to be published in the future, and our research team will review it.

Reviewer 2 Report

The manuscript entitled “Comparison of the malignant predictors in intrahepatic and extrahepatic intraductal papillary neoplasm of bile duct” evaluated the predictors of malignancy in E-IPNB and I-IPNB. The findings revealed E-IPNB has a higher rate of malignancy than I-IPNB. The predictive factor for malignancy is mural nodule > 12 mm in I-IPNB and mural nodule enhancement in E-IPNB. The manuscript is well written and presented and it is suitable for publication in JCM. There are a few comments that need to be revised:

  1. Some editing for the English language is required throughout the manuscript including typing errors (E.g. Heading for Table 4, 5... need to correct)
  2. Some recent articles need to include and discussion PMID: 33317146; https://ascopubs.org/doi/abs/10.1200/jco.2011.29.4_suppl.201;  PMID: 35141839
  3. Any invasive carcinoma need to mention and revise
  4. The conclusion needs to present in a more clear way since it means now generally. 
  5. Although many limitations were mentioned, no further suggestions or further works are mentioned. Also, the aim of the study needs to write more clearly. 

Author Response

The manuscript entitled “Comparison of the malignant predictors in intrahepatic and extrahepatic intraductal papillary neoplasm of bile duct” evaluated the predictors of malignancy in E-IPNB and I-IPNB. The findings revealed E-IPNB has a higher rate of malignancy than I-IPNB. The predictive factor for malignancy is mural nodule > 12 mm in I-IPNB and mural nodule enhancement in E-IPNB. The manuscript is well written and presented and it is suitable for publication in JCM. There are a few comments that need to be revised:

  1. Some editing for the English language is required throughout the manuscript including typing errors (E.g. Heading for Table 4, 5... need to correct)

Reply: Thank you for your comment. English proofreading was performed again through a proofreading company.

  1. Some recent articles need to include and discussion PMID:; https://ascopubs.org/doi/abs/10.1200/jco.2011.29.4_suppl.201;  PMID: 35141839

Reply: Thank you for your comment. We added those references as you suggested and added the sentence below in the discussion.

Page 9, line 22-26 “CA19-9 is not an independent predictor in our results, however, a recent study showed that CA19-9 is a prognostic marker of I-IPNB with high-grade dysplasia and carcinoma.[13] Therefore, despite our results, it should be noted that CA19-9 is always important as prognostic markers of the biliary tract cancer. 35141839

Page 9, libe 48-51 “Mutations expressed according to type 1/2 have also been reported to be different. In type 1, KRAS, GNAS, and RNF43 mutations were identified and reported to be similar to the intestinal subtype of IPMN. In type 2, mutations in TP53, SMAD4, and PIK3CA were identified.[20]” 33317146

  1. Any invasive carcinoma need to mention and revise

Reply: Thank you for your comment. We edited the manuscript per your comment.

Page 2, Line 78-79 IPNB associated with invasive carcinoma has stromal invasion in IPNB.

  1. The conclusion needs to present in a more clear way since it means now generally. 

Reply: Thank you for your comment. Per your comment, to make the conclusion clearer, we deleted one sentence.

Page 10, Of the patients with I-IPNB and E-IPNB, 53.4 % % and 76.3% were diagnosed with malignancy, respectively

We also added “with mucosal dysplasia” in the sentence below.

Page 10, Line 82-83 “In addition, 43.7% of non-operative patients with mucosal dysplasia had malignant transformation within three years.”

  1. Although many limitations were mentioned, no further suggestions or further works are mentioned. Also, the aim of the study needs to write more clearly. 

Reply: Thank you for your comment. We added more suggestions in the limitations section.

Page 10, line 77-79 “More studies should be published on IPNB, including on its natural course, histological results, and mutations in the future.”